# Electron cryo-tomography reveals the subcellular architecture of growing axons in human brain organoids

Patrick C Hoffmann[1†§], Stefano L Giandomenico[1†#], Iva Ganeva[1], Michael R Wozny[1], Magdalena Sutcliffe[1], Madeline A Lancaster[1*‡], Wanda Kukulski[1,2*‡]

[1]MRC Laboratory of Molecular Biology, Francis Crick Avenue, Cambridge, United Kingdom; [2]Institute of Biochemistry and Molecular Medicine, University of Bern, Bern, Switzerland

*For correspondence:
mlancast@mrc-lmb.cam.ac.uk
(MAL);
wanda.kukulski@ibmm.unibe.
ch (WK)

†These authors contributed equally to this work

‡These authors also contributed equally to this work

Present address: §Department of Molecular Sociology, Max Planck Institute of Biophysics, Frankfurt am Main, Germany; #Max Planck Institute for Brain Research, Frankfurt am Main, Germany

**Abstract** During brain development, axons must extend over great distances in a relatively short amount of time. How the subcellular architecture of the growing axon sustains the requirements for such rapid build-up of cellular constituents has remained elusive. Human axons have been particularly poorly accessible to imaging at high resolution in a near-native context. Here, we present a method that combines cryo-correlative light microscopy and electron tomography with human cerebral organoid technology to visualize growing axon tracts. Our data reveal a wealth of structural details on the arrangement of macromolecules, cytoskeletal components, and organelles in elongating axon shafts. In particular, the intricate shape of the endoplasmic reticulum is consistent with its role in fulfilling the high demand for lipid biosynthesis to support growth. Furthermore, the scarcity of ribosomes within the growing shaft suggests limited translational competence during expansion of this compartment. These findings establish our approach as a powerful resource for investigating the ultrastructure of defined neuronal compartments.

## Introduction

Mammalian neurons are uniquely specialized cells. With an elaborate tree-like structure and elongated shape, sometimes up to a meter in length in adult humans, they have the highest surface area to volume ratio of any cell in the body (*Winkle and Gupton, 2016*). Neurons are also extremely polarized, with structurally and functionally distinct axonal and dendritic compartments (*Bartlett and Banker, 1984*; *Bray, 1973*). While dendrites receive signals, the axon transmits the electrical signal to downstream targets. It is this directional relay, together with the specific topology of axonal connections and their bundling in defined tracts, that allows for information transfer, and ultimately cognition. The formation of such long-range tracts depends upon the coordinated growth of thousands and even millions of immature axons, often over great distances.

How the axon achieves such remarkable elongation remains to be determined, but its subcellular architecture is likely to reflect unique requirements for growth. For example, the organelle composition of the developing axon must be such that the neuron, without going through cell division, can undergo size increase rates akin to rapidly dividing cancerous cells (*Smith, 2009*). How are cellular compartments and the cytoskeleton organized within the rapidly extending axon? Does the subcellular organization provide the means to understand the supply of lipids and proteins, necessary for the increase in axon surface area during growth? These are some of the outstanding questions that nanometer-scale insights could help to address.

The cellular organization of different neuronal compartments, including the axon, has been studied by electron microscopy (EM) since its earliest days on fixed nervous tissue from rodents (*Palay, 1956*; *Palay and Palade, 1955*). Volumetric EM techniques have enabled 3D reconstructions of the arrangement of cells within native nervous tissue (*White et al., 1986*; *Ohyama et al., 2015*; *Wanner et al., 2016*; *Zheng et al., 2018*). Such large-scale approaches provide 'connectomics' information on the nervous systems of small animals such as nematodes, fly, and fish larvae. Recently, a study of subcellular details in fixed post-mortem human brain slices made use of correlative light and electron microscopy (CLEM) and electron tomography (ET) (*Shahmoradian et al., 2019*). Although these studies revealed many ultrastructural features, the resolution and interpretability of classical preparation methods are restricted by limited sample preservation.

In comparison to chemical fixation, high-pressure freezing followed by freeze-substitution was shown to result in improved preservation of mouse nervous tissue, highlighting the importance of cryo-fixation for a more accurate interpretation of ultrastructural details (*Korogod et al., 2015*). The combination of cryo-fixation with EM imaging at cryogenic temperatures is particularly suited to preserve the fine structure of neuronal cells (*Lučić et al., 2007*; *Fernández-Busnadiego et al., 2010*; *Zuber et al., 2005*). Furthermore, recent methodological advances in cryo-ET and subtomogram averaging have enabled in situ structural studies of protein complexes within their cellular context (*O'Reilly et al., 2020*; *Tegunov et al., 2021*; *Watanabe et al., 2020*; *Allegretti et al., 2020*). These methods can be applied to mammalian neuronal cell cultures to provide unprecedented views of their macromolecular architecture (*Guo et al., 2018*; *Bäuerlein et al., 2017*; *Trinkaus et al., 2021*; *Liu et al., 2020*). Nevertheless, it remains challenging to unequivocally identify axons in primary neuronal cultures (*Bartlett and Banker, 1984*; *Bray, 1973*) because the cells are dissociated from their tissue context, and therefore exhibit intermixed axons and dendrites. Furthermore, ultrastructural analysis of human axons within the near-native context of cryo-preserved tracts would be invaluable for understanding neurodevelopment, nerve damage, and regeneration (*Blanquie and Bradke, 2018*).

Cerebral organoids provide the opportunity to study aspects of human neuronal physiology within the context of a complex 3D cellular milieu mimicking the architecture of the developing brain (*Birey et al., 2017*; *Lancaster et al., 2017*; *Quadrato et al., 2017*). Recently, we developed an optimized air-liquid interface slice culture paradigm (*Giandomenico et al., 2019*) that enables long-term culture of cerebral organoids and promotes the establishment of axon tracts, consisting of dozens or, in thicker tracts, even up to thousands of individual axons bundled together, able to form functional connections. While the cell bodies are part of the organoid tissue, corticofugal (deep-layer identity) axon tracts project over long distances of several millimeters from the organoid (*Giandomenico et al., 2019*), effectively segregating the axons from dendrites and somata. This feature, combined with the ability to derive cerebral organoids from human cells, makes this method a promising route to examining the cell biology and molecular structure of human axons.

Here, we present a workflow that combines air-liquid interface cerebral organoid (ALI-CO) culture with cryo-CLEM to study the subcellular architecture of developing human axons in a context closely mimicking in vivo. Cryo-CLEM offers unambiguous identification of fluorescent cells, or individual cellular compartments, combined with high-resolution cryo-EM imaging of the same, near-natively preserved sample to reveal its underlying molecular structure (*Lučič et al., 2013*). We adapted this technology to observe the behavior of growing human axons within tracts in real time followed by electron cryo-tomography (cryo-ET) of the same axons with high structural preservation.

## Results

We first examined in detail axon growth behaviors of ALI-COs. Thick tracts that exited the ALI-COs stained positive for the pan-axonal marker SMI312 and negative for the dendritic marker MAP2 (*Figure 1A* and *Figure 1—figure supplement 1*). This, together with their length and physical distance from neuronal cell bodies within the organoid, consolidated that these tracts comprise almost exclusively axons (*Figure 1A* and *Figure 1—figure supplement 1*). To monitor growth, we next expressed a farnesylated membrane-targeted GFP (fGFP) (*Lancaster et al., 2017*; *Giandomenico et al., 2019*; *Figure 1—figure supplement 2*) in a subset of cells within the organoid slices. As a result, tracts contained a mixture of unlabeled and fluorescently labeled axons, which extended rapidly within the tracts (*Figure 1B* and *Figure 1—video 1*). We measured the extension speed for individual fluorescently labeled axons over several hours to be an average of 691 nm/min, with a peak pace of nearly

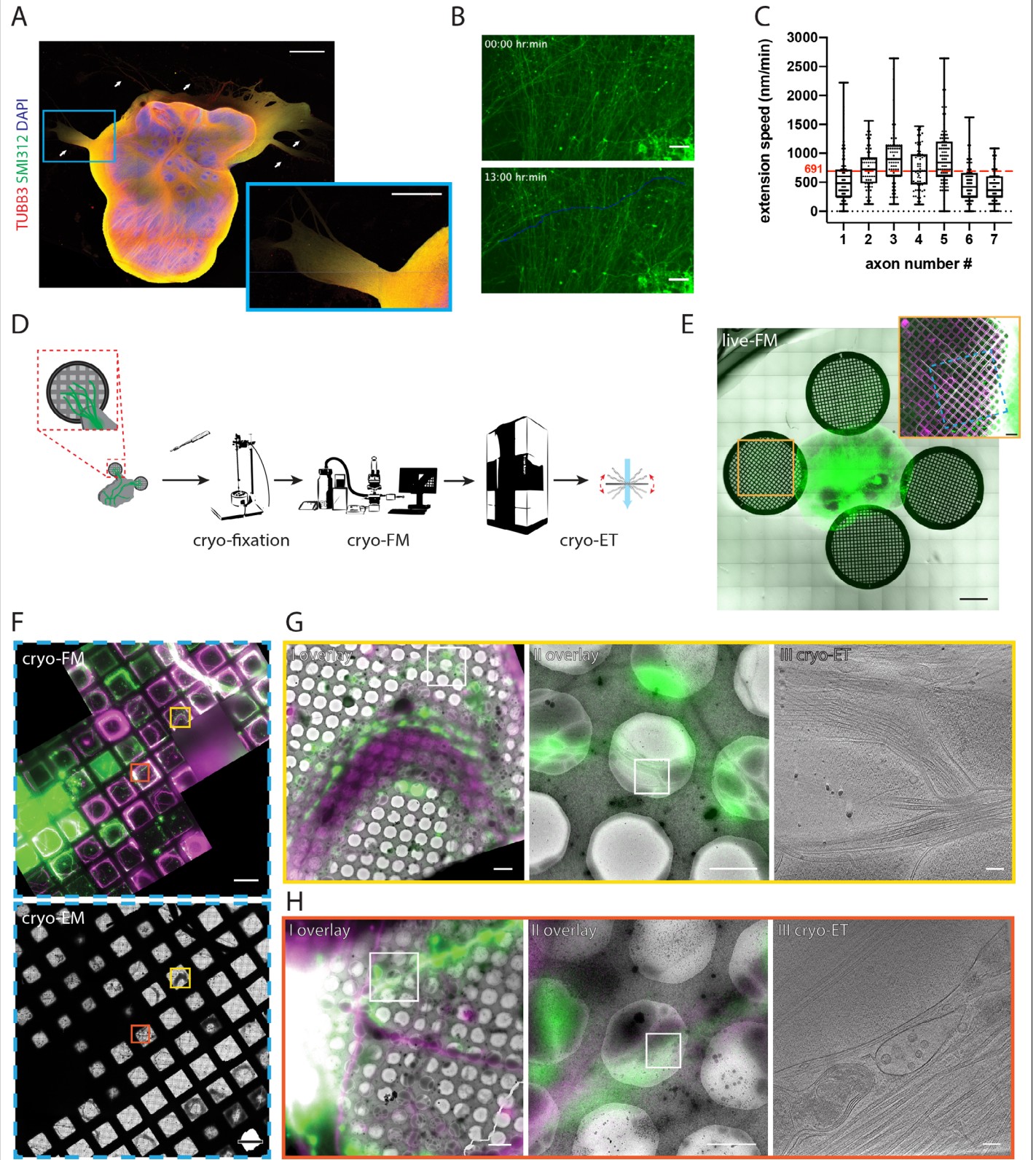

**Figure 1.** Targeting fluorescently labeled axons within tracts from human cerebral organoids by correlative light and electron cryo-microscopy (cryo-CLEM). (**A**) Immunofluorescence of an air-liquid interface cerebral organoid (ALI-CO) stained for the pan-neuronal marker TUBB3 and the pan-axonal marker SMI312. The box highlights a magnified view of escaping axon tracts. White arrows point to bundles of varying thickness. (**B**) Representative fluorescence microscopy (FM) images of extending fGFP⁺ expressing axons. The top panel is the first frame and the bottom panel is the last frame of

*Figure 1 continued on next page*

*Figure 1 continued*

the 13 hr-long live FM *Figure 1—video 1*. The images were used to track the axon labeled as number four in C., and the blue line marks the trajectory of its growth throughout the movie. (**C**) Box and whisker plot of axon extension speed measurements reporting the median, first and third quartiles, minimum and maximum. The individual data points represent the extension speed calculated between two consecutive frames. The measurements were done on 7 axons from three different ALI-COs. The red dashed line indicates the average speed of 691 nm/min, calculated from the shown 7 axons. (**D**) Schematic preparation of ALI-COs for cryo-CLEM including electron cryo-tomography (cryo-ET). (**E**) Overlay of fluorescence (GFP) and transmitted light overview images of grids placed around an ALI-CO. Lower magnification image shows day 1. Inset shows axons labeled with fGFP and additional tracts stained with SiR-tubulin immediately prior to cryo-fixation after 11 days of growth on the grid to the left of the ALI-CO, indicated by the orange square. (**F**) Cryo-FM (top) and cryo-EM (bottom) overview of the blue area indicated on the grid shown in the inset in E. Yellow and red boxes indicate the areas shown in G. and H., respectively. (**G** and **H**) Correlated cryo-FM and cryo-ET on two different grid squares. Subpanels I and II: Overlays of cryo-FM and cryo-EM at different zoom levels to identify individual fluorescent axons. Areas in subpanels II correspond to white squares in subpanels I. Subpanels III: Virtual slices through cryo-tomograms acquired at the positions of white squares indicated in subpanels II. Scale bars: 1 mm and 500 µm (inset) in A., 50 µm in B., 1 mm and 200 µm (inset) in E., 100 µm in F., 5 µm (I), 2 µm (II) and 100 nm (III) in G. and H. GFP is shown in green, SIRtubulin in magenta in E – H.

The online version of this article includes the following video and figure supplement(s) for figure 1:

**Figure supplement 1.** ALI-CO escaping tracts comprise principally axons.

**Figure supplement 2.** Plasmid constructs and electroporation strategy employed in the study.

**Figure 1—video 1.** Time-lapse movie of the extending fGFP⁺ axon shown in Figure 1B.

https://elifesciences.org/articles/70269/figures#fig1video1

**Figure 1—video 2.** Time-lapse movie over 10 days of fGFP⁺ axon tracts growing on an EM grid.

https://elifesciences.org/articles/70269/figures#fig1video2

3000 nm/min (*Figure 1C*). The average speed we observed is comparable to previous reports of chicken dorsal-root ganglia (*Ketschek et al., 2007*). Intrigued by the dynamicity of axon pathfinding, we sought to establish an approach that would allow us to examine the subcellular organization that underlies this cellular behavior.

For this, we developed a procedure to prepare axon tracts for cryo-EM (*Figure 1D*). We placed coated EM support grids in close proximity to ALI-COs on organotypic cell culture inserts (*Figure 1E*). The signal of fGFP⁺ axons was used to track the behavior of axons over time by live fluorescence microscopy (live-FM). We monitored their growth over the course of 11 days (*Figure 1E* and *Figure 1— video 2*). Prior to cryo-fixation, in a subset of experiments we applied SiR-tubulin to visualize all tracts on the grid (*Figure 1E* inset). Grids on which axon tracts approached the center were detached from the organoid slice using a biopsy punch and immediately cryo-fixed by plunge-freezing (*Figure 1D*). Detachment and plunge-freezing were coordinated between two experimenters who sequentially handled the grids, to keep the time from detachment until completed cryo-fixation as short as possible and within less than 20 s (see Materials and methods). To specifically target fGFP⁺ axons within tracts by cryo-ET we used cryo-CLEM, imaging each grid both by cryo-fluorescence microscopy (cryo-FM) and cryo-EM (*Figure 1F*). Fluorescent signals of individual tracts could be correlated to cryo-EM overviews by using landmark features on the grid (*Figure 1G,H,I*). This allowed us to distinguish fGFP⁺ axons within the tracts from axons that were positive only for SiR-tubulin (*Figure 1G, H,II*). We then imaged fGFP⁺ axons by cryo-ET, targeting specific segments of individual axons within axon tracts (*Figure 1G, H,III*). This approach allowed us to establish a direct link between cellular behavior observed live and cellular ultrastructure (*Figure 2—video 1*).

In our cryo-ET data, cellular structures such as protein assemblies and membranes were preserved to a high level of detail (*Figure 2A*). We observed unbranched, longitudinally aligned actin filaments, recognizable by their characteristic thickness of 7–9 nm and an apparent pitch of 5–6 nm (*Egelman et al., 1982*; *Figure 2A*, orange arrows). Microtubules revealed their individual protofilaments (*Figure 2A*, magenta arrows), as well as numerous intraluminal protein densities (*Garvalov et al., 2006*; *Foster et al., 2021a*). Microtubule bundles were often so dense as to seemingly pose constraints on microtubule-based transport. We observed mitochondria within these bundles (*Figure 2A*, green arrows), suggesting that dynamic rearrangements are required to allow sufficient space for vesicles and organelles to pass. We also found filaments of about 10 nm in diameter (*Figure 2A*, brown arrows). These dimensions match those of neurofilaments (NFs) such as NF-L, NF-M, and NF-H (*Malka-Gibor et al., 2017*), which serve as signature markers for axons. Our tomograms also revealed coated and uncoated vesicles (*Figure 2A*, yellow arrows, and *Figure 2—figure supplement 1A*). Furthermore,

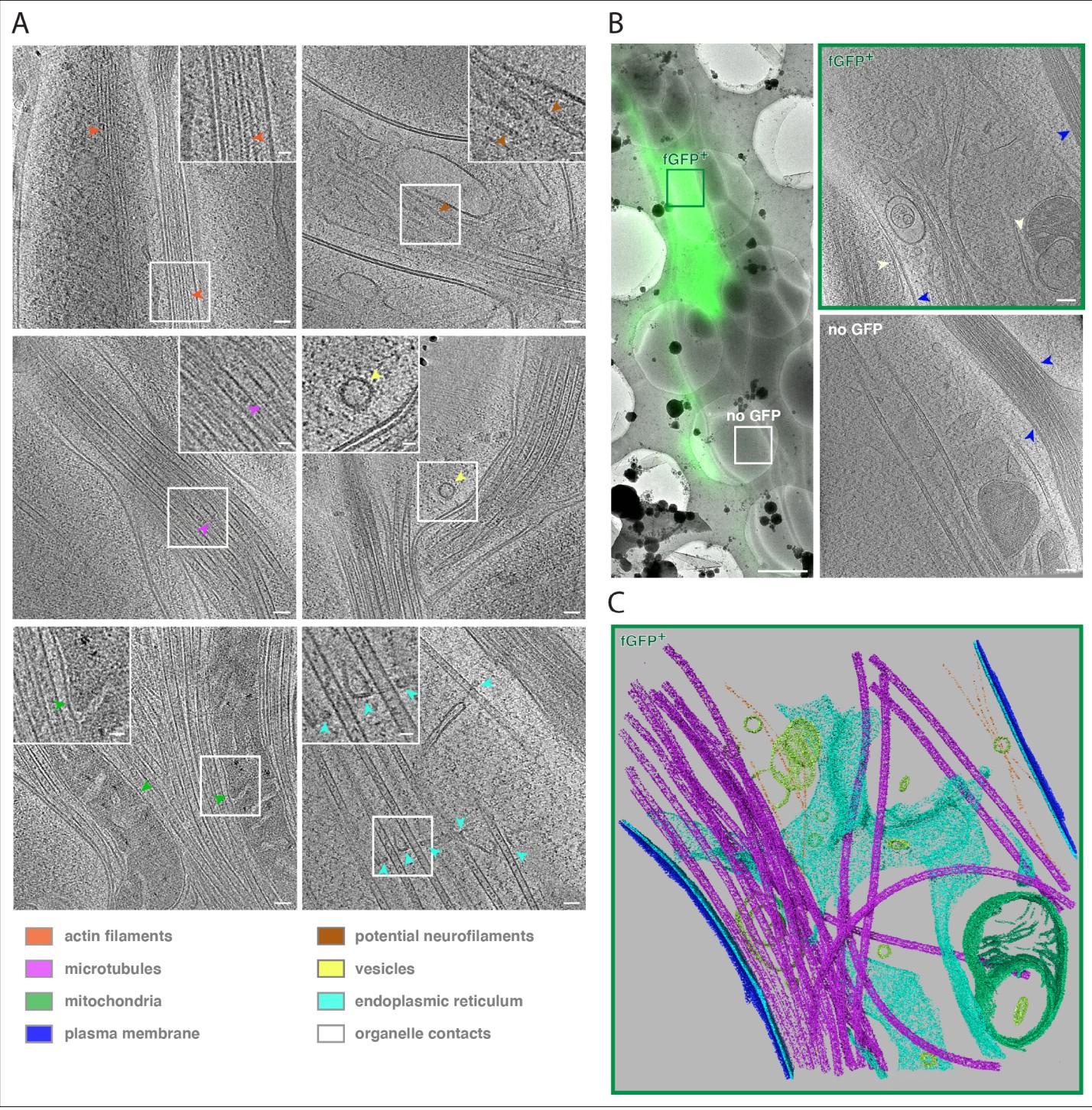

**Figure 2.** Cryo-ET reveals subcellular features of developing human axon tracts. (**A**) Cryo-tomograms reveal bundles of unbranched, longitudinally aligned actin filaments, recognizable by actin subunit arrangement (orange arrows), microtubules filled with lumenal densities, microtubule protofilaments (purple arrows), mitochondria (green arrows) embedded in microtubule bundles, potential neurofilaments (brown arrows), vesicles carrying protein cargo (yellow arrow), endoplasmic reticulum (cyan arrows) closely associated with microtubules. Insets show magnified views of the boxed areas. (**B**) fGFP+ and untransfected axons are found within the same axon tract, allowing direct phenotype comparison. Left panel: Overlay of cryo-FM and cryo-EM overview images. Two right panels: Virtual slices through cryo-tomograms of fGFP+ and control axon. White arrows indicate contacts between ER and mitochondria as well as between ER and plasma membrane, blue arrows indicate cell-cell contacts between different axons. (**C**) Segmentation model of the fGFP+ axon shown in B., illustrating the complexity of the cellular ultrastructure. Microtubules are shown in magenta, actin filaments in orange, endoplasmic reticulum in cyan, vesicles and other membrane compartments in yellow, mitochondrial membranes in green and

*Figure 2 continued on next page*

*Figure 2 continued*

the plasma membranes of the fGFP+ axon and neighboring axons in blue and dark blue, respectively. Scale bars: 50 nm (20 nm for insets) in A., 2 μm in B. left panel and 100 nm in B. right two panels.

The online version of this article includes the following video and figure supplement(s) for figure 2:

**Figure supplement 1.** Vesicles: Associations with the cytoskeleton and size distributions.

**Figure supplement 2.** Gallery of correlative cryo-FM and cryo-ET dataset from membrane targeted GFP (fGFP) expressing ALI-COs.

**Figure supplement 3.** Gallery of correlative cryo-FM and cryo-ET dataset from L1CAM-GFP expressing ALI-COs.

**Figure supplement 4.** Gallery of correlative cryo-FM and cryo-ET dataset from GFP-ESYT1 expressing ALI-COs.

**Figure supplement 5.** Cell-cell contacts in axons from fGFP and L1CAM-GFP expressing ALI-CO slices.

**Figure supplement 6.** ER-plasma membrane contact sites in axons from fGFP and GFP-ESYT1 expressing ALI-CO slices.

**Figure 2—video 1.** Correlative microscopy of axons from growing tracts across scales to the supramolecular architecture of individual axon shafts.
https://elifesciences.org/articles/70269/figures#fig2video1

**Figure 2—video 2.** Tomogram and segmentation of the fGFP+ axon shaft shown in Figure 2C.
https://elifesciences.org/articles/70269/figures#fig2video2

we frequently observed large expanses of endoplasmic reticulum (ER) cisternae, which extended into thin membrane tubules tightly associated with microtubules (*Figure 2A*, cyan arrows). We also observed membrane contact sites between ER-mitochondria and ER-plasma membrane (*Figure 2B*, white arrows).

Finally, as each tract contained several axons, our tomograms frequently visualized axon-axon contacts, a unique physiological feature of axons in their normal tissue context (*Figure 2B*, blue arrows). Because only a subset of cells expressed fGFP, most tracts contained a large proportion of unlabeled, GFP negative axons (*Figure 2B*, left panel). By this approach, GFP-negative axons could be imaged on the same EM grid (*Figure 2B*, right panels), simplifying the acquisition of control data as well as making comparison more reliable by removing grid-to-grid variability as a source of unspecific differences. We did not observe any obvious ultrastructural differences between fGFP+ and GFP negative axons (*Figure 2—figure supplement 2*). To highlight the ultrastructural complexity, we segmented key cellular elements of a fGFP+ axon volume (*Figure 2C*, bottom panel and *Figure 2—video 2*).

The 40 tomograms we collected comprised unlabeled axons as well as axons expressing either fGFP, GFP-tagged human L1 cell adhesion molecule (L1CAM) or GFP-tagged extended-synaptotagmin isoform 1 (ESYT1) (*Figure 1—figure supplement 2*, *Figure 2—figure supplements 2–4*). We overexpressed L1CAM, which is involved in axon-axon contacts (*Siegenthaler et al., 2015*), and ESYT1, involved in lipid metabolism (*Saheki et al., 2016*), as possible modulators of tract formation and axon elongation. Next, we analyzed potential changes in subcellular structure upon L1CAM-GFP or GFP-ESYT1 overexpression (*Figure 2—figure supplements 3 and 4*). We compared cell-cell contacts formed between individual axons in tracts from ALI-CO slices expressing fGFP with tracts from ALI-CO slices overexpressing the cell adhesion molecule L1CAM. While the separation between plasma membranes of adjacent axons within tracts was remarkably narrow, we found no significant difference between cell-cell contacts from fGFP and L1CAM-GFP overexpressing ALI-COs (fGFP: mean = 5.63 nm, SD = 0.65 nm, N = 25 and L1CAM-GFP: mean = 5.29 nm, SD = 0.58 nm, N = 9) (*Figure 2—figure supplement 5*). We also compared the occurrence of ER-plasma membrane contact sites in axons from ALI-CO slices expressing fGFP with axons from ALI-CO slices overexpressing the ER-plasma membrane contact site protein GFP-ESYT1 (*Fernández-Busnadiego et al., 2015*). The frequency with which we observed such contact sites was similar in both data sets (86 % of fGFP tomograms (N = 22) and 90 % of GFP-ESYT1 tomograms (N = 10) contained at least one instance in which the ER was within approximately 30 nm distance from the plasma membrane) (examples in *Figure 2—figure supplement 6*). We cannot exclude that there may be subtle unanticipated phenotypes associated with other, specific structures, nor that more severe phenotypes could be masked by the mosaic nature of the bundles, with wild-type axons possibly driving growth of axons expressing the transgene. Nonetheless, the intactness of the subcellular organization suggests that in principle our approach allows one to assess the impact of gene manipulation on cellular ultrastructure without technical knock-on effects.

We next set out to analyze the observed cellular structures in more detail. For that, we pooled the tomograms of the three data sets. First, we investigated the localization of vesicles (*Figure 2A* and *Figure 2—figure supplement 1A*). While most vesicles appeared free in the cytosol, approximately 1 7 % were associated with microtubules and 14 % with actin (N = 290, *Figure 2—figure supplement 1B*). The vesicles had mean diameters of about 50 nm (52.00 nm, SD = 19.42 nm, N = 200 for free vesicles; 50.92 nm, SD = 16.70 nm, N = 49 for MT associated vesicles, and 52.18 nm, SD = 17.84 nm, N = 41 for actin associated vesicles) (*Figure 2—figure supplement 1C*). These sizes are similar to vesicles in axons of mouse dorsal root ganglia neurons (*Foster et al., 2021b*). Although we did not determine the origin and identity of the vesicles, their distribution suggests that at least a subset could correspond to secretory vesicles (*Gumy et al., 2017*).

A defining feature of axon identity is the parallel arrangement of bundled microtubules (*Burton, 1988*; *Heidemann et al., 1981*). Therefore, we sought to determine the polarity of individual microtubules in tomograms using subtomogram averaging (*Figure 3A*). We used axial views of each microtubule average to determine the handedness and thus polarity of the microtubule based on the tilt of its protofilaments (*Sosa and Chrétien, 1998*; *Figure 3A and B*). This analysis showed that the majority of bundles had a uniform, parallel microtubule arrangement, further confirming axon identity (*Figure 3C*). Because it is difficult to trace the complete length of individual axons on the EM grid and growth was not necessarily uni-directional, we could not unambiguously determine the microtubule orientation relative to the axon leading edge. In total, we analyzed between three and nine microtubules in nine tomograms by subtomogram averaging. While the protofilament tilts could be determined unambiguously in a subset of five axon tomograms (*Figure 3C*), the number of protofilaments could be determined for 28 individual microtubules. In some cases, the close proximity to other microtubules in larger bundles and the anisotropic resolution of the tomographic data prevented analysis of the protofilament number. We found that the majority of microtubules consisted of 13 protofilaments, while 2 microtubules had 12 protofilaments (N = 28) (*Figure 3D and E*). Although we cannot exclude that the microtubule organization we observed was influenced or stabilized by labeling with SiR-tubulin in the subset of experiments where this labeling was performed, 13 protofilaments have been suggested before to be the predominant molecular architecture of microtubules in human cells (*Watanabe et al., 2020*; *Chaaban and Brouhard, 2017*). We calculated a 3D subtomogram average from 16 of the 13-protofilament microtubules, which revealed the typical 4 nm repeat of individual tubulin subunits along the protofilaments (*Figure 3—figure supplement 1*; *Amos and Klug, 1974*; *Nogales et al., 1999*).

The rapid growth of developing axon tracts must be supported by synthesis of new biomolecules. At average growth rates of 691 nm/min (*Figure 1C*), the demand of membrane and lipid molecules must be high. Using the plasma membrane as a proxy to estimate the amount of newly synthesized membrane, we segmented the plasma membrane in tomograms and estimated the surface area of individual axon shafts (*Figure 3F*, see also Materials and methods). Each micrometer of length on average corresponded to approximately $1.24 \times 10^6$ nm$^2$ of plasma membrane area (SD = $7.3 \times 10^5$ nm$^2$, N = 14) (*Figure 3G*). The broad distribution reflects the variability between the thinnest parts of axon shafts and axonal varicosities. Because each μm$^2$ of membrane contains approximately $5 \times 10^6$ lipids (*Alberts et al., 2002*), corresponding to five lipid molecules/nm$^2$, we estimated the lipid supply to the plasma membrane required to sustain the average growth rate of an axon as approximately $4.3 \times 10^6$ lipid molecules/min. This massive influx of new phospholipids into the plasma membrane points to a unique requirement for lipid biosynthesis and transfer.

The ER is the major organelle for lipid biosynthesis. We therefore examined ER ultrastructure to assess whether it may help explain how the unique lipid requirements of the growing axon are supported. Large flattened cisternae (*Figure 2B and C*) were reminiscent of the ER observed in cultured neurons (*Schrod et al., 2018*). We further found tubular segments of axonal ER that were remarkably narrow (*Figure 3H*). At their thinnest outer diameter, the ER tubules in axon shafts measured on average 10.8 nm (SD 2.0 nm, 82 measurements), which is narrower than previously observed in mouse axons (*Terasaki, 2018*). For comparison we measured the thinnest ER tubules in HeLa cells and found them to be about twice the diameter (20.4 nm, SD 3.2 nm, 24 measurements) (*Figure 3H,I*). Some of the axonal ER tubules had a local outer diameter of less than 10 nm. Considering that this measurement includes the bilayer thickness, this implies that these ER tubules of human axons contain hardly any lumenal space, likely posing constraints on diffusion of ER proteins. The observation that the ER

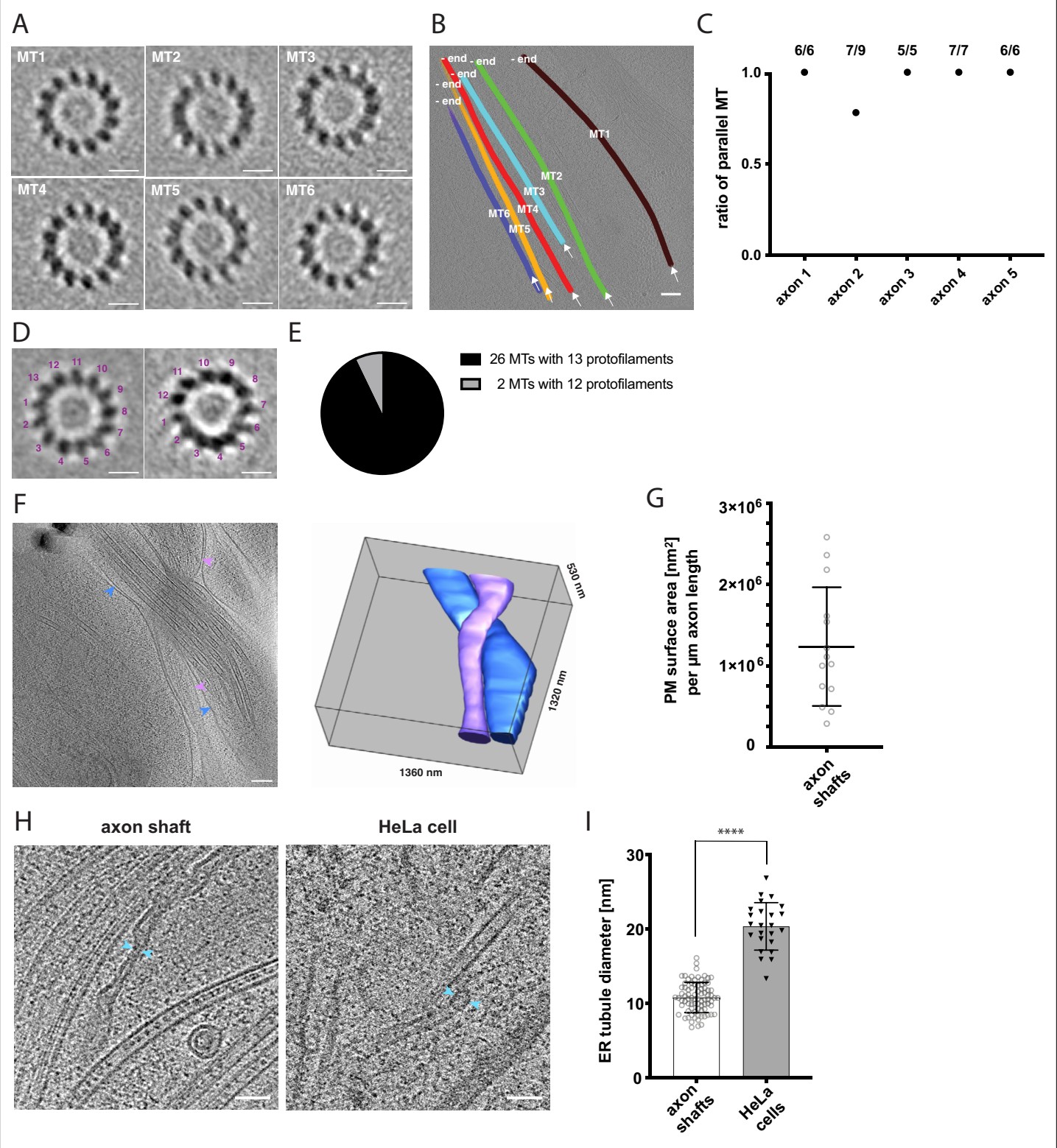

**Figure 3.** Cytoskeletal and membrane architecture of developing axon shafts. (**A**) Axial views of subtomogram averages reveal the polarity of each microtubule (MT) shown in B. The directionality of the microtubule was determined from the radial tilt of the protofilaments seen in axial views (*Sosa and Chrétien, 1998*). (**B**) Tomographic slice with six microtubules depicted as differently colored tubes (MT1-MT6) (same tomogram as shown in *Figure 2B*, 'no GFP' panel). Arrows indicate the viewing direction of the axial views of subtomogram averages shown in A. The -end of each microtubule is indicated, determined by the analysis shown in A. (**C**) Ratio of parallel microtubules determined by subtomogram averaging in five different axons

*Figure 3 continued on next page*

*Figure 3 continued*

(6–9 individual microtubules per axon). Axon one is depicted in A. and B. (**D**) Axial view on subtomogram average examples of individual microtubules with 13 protofilaments (left) and with 12 protofilaments (right). (**E**) Occurrence of microtubules with 12 and 13 protofilaments, determined by individual subtomogram averaging of 28 microtubules as shown in D. (**F**) Virtual tomographic slice and the corresponding segmentation model of the plasma membrane of two individual axon shafts. Blue and magenta arrows indicate plasma membrane segments visible in the virtual tomographic slice. (**G**) Plasma membrane (PM) surface area measurements in $nm^2$ normalized to the length in $\mu m$ of 14 axon shafts captured in 12 tomograms. (**H**) The curvature of ER tubules is higher in axon shafts than in HeLa cells. Arrows indicate the shortest distance between the membrane bilayer cross sections. (**I**) Diameters of the thinnest ER tubules measured in axon shafts and in HeLa cells (82 and 24 measurements, respectively). Welch's t test was employed for statistical analysis: $p < 0.0001$. Scale bars: 10 nm in A. and D., 100 nm in B. and F., 50 nm in H.

The online version of this article includes the following figure supplement(s) for figure 3:

**Figure supplement 1.** Subtomogram average of 13 protofilament microtubules.

is depleted of lumen whilst adopting highly curved tubular shapes indicates high local membrane surface-to-ER volume ratios, and suggests that the axonal ER structure may be a consequence of maximized synthesis of lipids, produced to sustain high growth rates during axon lengthening.

We anticipated the extending axon to require not only lipids but also proteins for maintaining functionality during elongation. We thus examined the presence of protein synthesis machinery in the elongating axon shafts. While it is known that mature axons do not display extensive Nissl bodies, indicating low ribosomal RNA content (*Angevine, 2002*), we reasoned that developing axons, due to their growth state, could have different protein biosynthesis requirements and hence composition. Furthermore, cellular cryo-ET provides the resolution to detect ribosomes, and to even distinguish between monosomes and polysomes (*Brandt et al., 2010*), making it a powerful method for direct detection of the protein synthesis machinery. It was therefore noteworthy that we observed a scarcity of potential ribosomes within our tomograms of growing axon shafts. In contrast, we readily identified a large number of ribosomes based on their size, shape and high contrast in tomograms of HeLa cells (*Ader et al., 2019*) and of other neuronal processes (*Figure 4A*). In order to assess the occurrence of ribosomes quantitatively, we counted ribosome-like particles, both cytosolic and membrane-bound, and found on average two particles per $\mu m^3$ in axon shafts (SD = 3 particles, n = 31 tomograms), 589 particles per $\mu m^3$ in other neuronal processes (SD = 344 particles, n = 4 tomograms), and 2314 particles per $\mu m^3$ in HeLa cells (SD = 977 particles, n = 5 tomograms) (*Figure 4B* and *Figure 4—figure supplement 1A*). The local ribosome concentration in axon shafts is thus less than 3.5 nM, about 1000-fold lower than in HeLa cells. The ribosome-like particles found in tomograms of axon shafts had an average diameter of 26.8 nm (SD = 3.7, N = 27), in agreement with the dimensions of human ribosomes (*Figure 4—figure supplement 1B*; *Anger et al., 2013*). 11 of them were in close proximity to ER membrane (*Figure 4—figure supplement 1C*). Thus, the large amounts of axonal ER had a minute number of ribosomes attached, supporting the idea that the ER has a primary function in lipid metabolism rather than protein synthesis. To validate these findings by an approach that would allow analysis of axons as well as dendrites within the same preparation, we tested for the presence of five distinct ribosomal proteins in neurons from dissociated organoids by immunofluorescence (*Figure 4C* and *Figure 4—figure supplement 2*). In agreement with the cryo-ET data, axon shafts identified as SMI312[+]/MAP2[-] showed significantly lower ribosomal signal than dendrites (SMI312[-]/MAP2[+]) (*Figure 4D*).

## Discussion

To date, cryo-ET studies of neurons have been conducted on rodent primary cells grown on EM grids (*Tao et al., 2018*; *Garvalov et al., 2006*; *Lučić et al., 2007*). Such dissociated neurons exhibit a random pattern of neurites, where axons typically grow along dendrites and do not form the directed long-range tracts seen in the context of tissue. Additionally, the growth state of axons and their developmental stage (i.e. axon pathfinding vs. synaptogenesis) in dissociated primary cultures are not always clear. Moreover, cryo-EM of human neurons has so far remained out of reach. We demonstrate that these limitations can be overcome by the combination of human cerebral organoids with cryo-CLEM. Because this approach requires axons to extend from the edge to the center of the grids, most of the analyzed cell areas corresponded to relatively mature pathfinding axons, which had elongated for 1–2 mm over the course of 10–14 days, during which the axon initial segment, shaft and growth

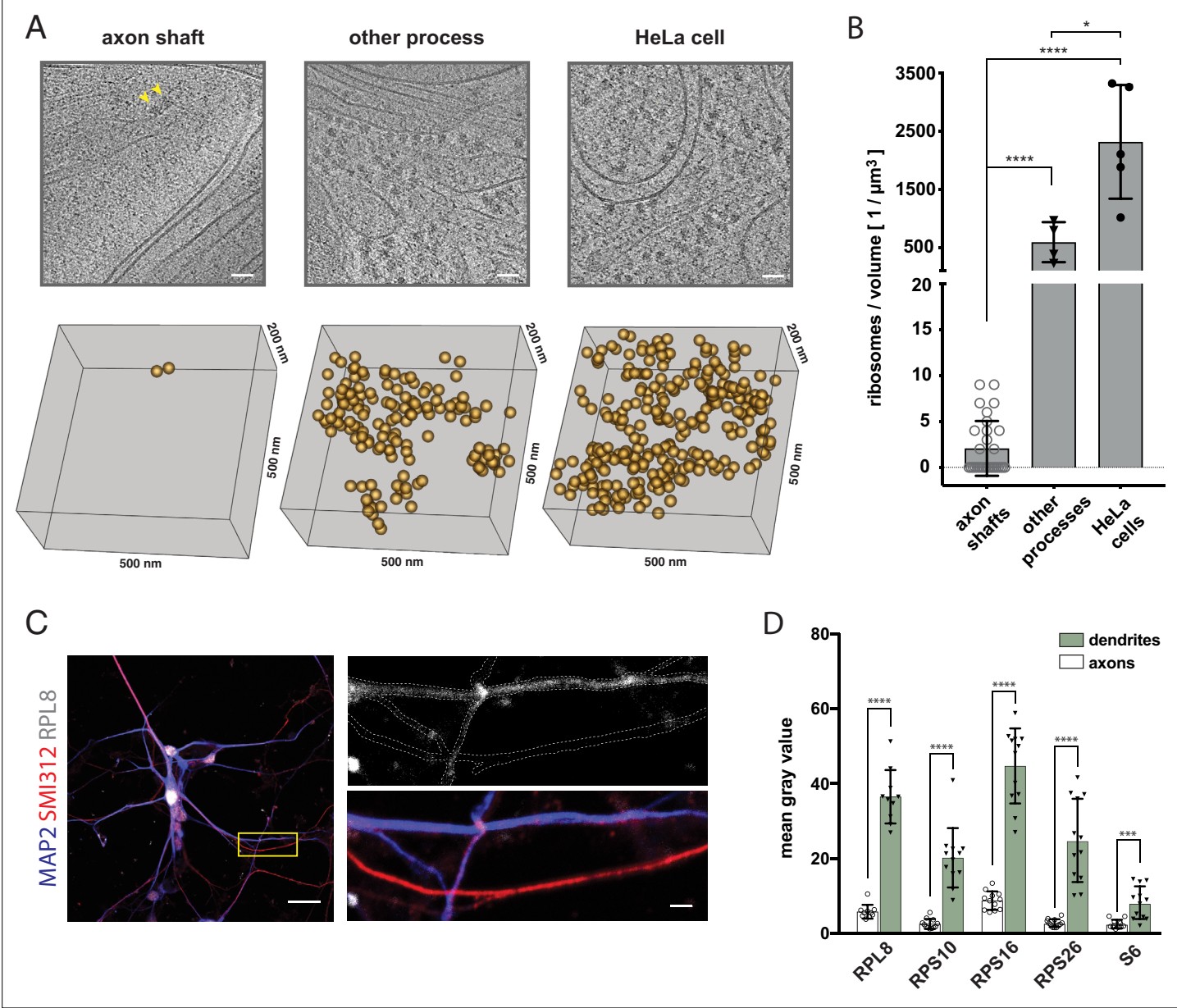

**Figure 4.** The shaft of the growing axon is scarcely populated by ribosomes. (**A**) Examples of ribosomes observed in cryo-tomograms of axon shafts from ALI-COs, of other cellular processes from ALI-COs, and of HeLa cells. The bottom panel shows 0.05 μm³ cryo-ET volumes, corresponding to the area shown in the upper panel. Positions of all ribosome-like particles observed in that volume are shown as orange spheres. (**B**) Comparison of the numbers of ribosome-like particles, normalized to the tomographic volume, observed in axon shafts, other processes and HeLa cells. Individual data points represent individual cryo-tomograms (30, 4, and 5 tomograms, respectively). Mann-Whitney tests were employed for statistical analysis: $p < 0.0001$ (****); $p < 0.05$(*). (**C**) Immunofluorescence images of dissociated neurons from organoids reveal low signal for the ribosomal 60 S component RPL8 in axon processes (identified by SMI312+/MAP2- labeling) in comparison to dendrites (identified by SMI312-/MAP2+ labeling). The yellow box outlines the area magnified in the right panel. The top image of the right panel shows the immunofluorescence signal for the ribosomal subunit RPL8. The bottom image shows the SMI312/MAP2/RPL8 composite. The white dashed line depicts the outline of axons and dendrites and was traced based on the MAP2 and SMI312 signal. The image shown is representative of the data used for quantifications shown in D. (**D**) Quantification of immunofluorescence images of ALI-CO derived dissociated neurons labeled for five distinct ribosomal proteins. The bars report the mean pixel grey value along axons and dendrites (mean ± SD). Each data point represents a different axon or dendrite. With the exception of quantifications done on the ribosomal protein S6, axons were identified as SMI312+/MAP2- neuronal processes, while dendrites were identified as SMI312-/MAP2+ neuronal processes. Due to antibody incompatibility, in the case of S6, dendrites were identified as MAP2+ processes while axons were identified as GFP+/MAP2- processes. The data pertains to one biological replicate. Mann-Whitney tests were employed for statistical analysis: RPL8, N = 10, $p < 0.0001$ (****); RPS10, N = 12, $p < 0.0001$ (****); RPS16, N = 12, $p < 0.0001$ (****); RPS26, N = 12, $p < 0.0001$ (****); S6, N = 12, $p = 0.0001$ (***). Scale bars: 50 nm in A., 20 μm and 2 μm in C.

*Figure 4 continued on next page*

*Figure 4 continued*

The online version of this article includes the following figure supplement(s) for figure 4:

**Figure supplement 1.** Ribosome-like particles in developing human axon shafts.

**Figure supplement 2.** Immunofluorescence analysis of axons and dendrites supports ribosomal depletion observed by cryo-ET in the axon shaft.

cones have established. Although we cannot exclude that a fraction of the regions imaged by cryo-ET were in the process of retraction or corresponded to axon branches, their position near the grid center indicates they had undergone axonal outgrowth. Thus, this approach allows for visualization of the intracellular landscape of axon shafts, an understudied region of actively growing human axons, unambiguously identified and in an environment that mimics the physiological context of an axon tract.

Our method of combining organoid technology with cryo-ET enabled us to visualize architectural characteristics of growing axon shafts that can help to explain their unique behaviors. The organization of the cytoskeleton, in particular the dense microtubule bundles, likely reflects the importance of mechanical properties, such as rigidity, and of cytoskeletal transport for projection of axons over long distances. Growing axon shafts also appear to contain exclusively smooth ER, similarly to mature nervous tissue depicted in textbook electron micrographs. We observed an intricate ER morphology with extremely narrow tubules, which were half the diameter previously observed in mature mouse axons by serial-section EM (*Terasaki, 2018*). Such ER morphology with minimal ER lumen volume suggests a role primarily in lipid biosynthesis to provide membrane material. In addition, we observed membrane contact sites between the ER and the plasma membrane, likely representing sites for transfer of newly synthesized lipid molecules, as lipid flux is a major function of membrane contact sites (*Levine, 2004*; *Prinz, 2014*). Furthermore, the tight association of the ER with microtubules could reflect attachment needed to pull the ER along the extending axon. Alternatively, this arrangement could represent piercing of ER cisternae by microtubules that grow quickly along the extending axon. These structural features of axonal ER may explain how axons acquire their unique surface area-to-volume ratio through lipid supply during elongation.

Quantification of ribosome occurrence in the axon shaft provides an estimate of the potential for local protein biosynthesis. Previous EM studies have reported clusters of ribosomes in the axon initial segment and growth cones (*Bartlett and Banker, 1984*; *Tennyson, 1970*; *Yamada et al., 1971*; *Bunge, 1973*; *Steward and Ribak, 1986*; *Koenig et al., 2000*). Mature presynaptic termini were shown to harbor ribosomal components, consistent with the presence of a distal ribosome pool, and metabolic labeling experiments demonstrated presynaptic protein synthesis (*Hafner et al., 2019*; *Shigeoka et al., 2016*). Within the extending axon shaft, the presence of an active pool of ribosomes is less clear. In contrast to the known hallmarks for identifying pre- and post-synaptic compartments, axon shafts are not easily identified through defining structural features; they can be mistaken for thin dendrites and therefore their ultrastructural features, especially during development, have remained poorly described. Our work demonstrates that the experimental setup described here allows one to confidently ascribe observations to axon shafts and, in principle, other axonal compartments within a neuronal tissue context. Our findings reveal that the shafts of developing axons specifically exhibit low ribosome numbers. Thus, in the early stages of neurodevelopment, during axon pathfinding, the axon shaft differs from dendrites in that it displays reduced ribosome levels.

These observations complement previous findings of local translation at growth cones and presynaptic termini, and suggest that different regions of the axon can exhibit very different supramolecular landscapes. We therefore propose a refined model in which biological processes in the axon initial segment, the growth cone, and presynaptic termini rely on local translation (*Bartlett and Banker, 1984*; *Yamada et al., 1971*; *Tennyson, 1970*; *Bunge, 1973*; *Steward and Ribak, 1986*; *Koenig et al., 2000*; *Hafner et al., 2019*; *Shigeoka et al., 2016*), while the axon shaft during growth is depleted of translational machinery and thus displays limited capacity for new protein synthesis. Furthermore, because protein synthesis in the growth cone was previously shown to be used primarily for navigation and not for extension (*Campbell and Holt, 2001*), ribosomal depletion along the shaft might represent a mechanism to desensitize this structure to exogenous guidance cues and ensure correct wiring.

Overall, these findings demonstrate that the combination of cryo-CLEM and human cerebral organoid technology is a powerful method to address the cell biology and architecture of its

compartments. Conceptually this approach is not limited to axons and could be extended to other organoid-derived cell types as long as the preparation allows sample vitrification. For instance, this methodology could potentially be applied to blood vessel organoids (*Wimmer et al., 2019*) to study ultrastructural aspects of vasculature development and vasculopathy, since, like neurons, the developing vasculature extends long, thin structures that would be amenable to vitrification in the same manner. Furthermore, a declination of this approach could take advantage of the organoid system to generate defined cell types difficult to isolate from primary samples. For example, one could place neural crest organoids (*Foltz et al., 2021*) in close proximity to EM grids to capture neural crest cells and study their ultrastructure during migration and differentiation into different neural crest cell derivatives. Such an approach, in combination with thinning of the cells by cryo-focussed ion beam milling (*Mahamid et al., 2016*), would provide a way to perform cryo-EM studies on physiologically relevant migratory cell types.

In conclusion, the combination of cerebral organoid technology and cryo-ET provides a window into the architecture of the human axonal compartment. This experimental setup could be extended to other organoid systems. We believe this methodology holds the potential to further our understanding of neuronal development, neurodegeneration and nerve injury.

## Materials and methods

**Key resources table**

| Reagent type (species) or resource | Designation | Source or reference | Identifiers | Additional information |
|---|---|---|---|---|
| Cell line (*H. sapiens*) | H9 hESC | DOI:10.1126/science.282.5391.1145 | hPSCReg ID: WAe009-A | |
| Cell line (*H. sapiens*) | HeLa control | other | | Cell line stably expressing Fsp27-GFP from tet-inducible promoter. Obtained from Koini Lim (lab of David Savage). |
| Recombinant DNA reagent | pCAGEN-SB100X | DOI:10.1038/nbt.3906 | | |
| Recombinant DNA reagent | pT2-CAG-fGFP | DOI:10.1038/nbt.3906 | | |
| Recombinant DNA reagent | pT2-CAG-fFusionRed | DOI: 10.1038/s41593-019-0350-2 | | |
| Recombinant DNA reagent | EGFP-E-Syt1 | Addgene | 66830 | |
| Recombinant DNA reagent | pcDNA3-hL1 | Addgene | 89411 | |
| Recombinant DNA reagent | pT2-CAG-GFP-E-Syt1 | Generated for this study | | |
| Recombinant DNA reagent | pT2-CAG-hL1CAM-GFP | Generated for this study | | |
| Other | Gemini X2 HT | BTX Harvard Apparatus | 45–2041 | |
| Other | Petri Dish Platinum Electrode for Tissue Chamber Kit 5 mm | BTX Harvard Apparatus | 45–0505 | |
| Other | Leica VT1000 S | Leica Biosystems | | |
| Commercial assay or kit | STEMdiff Cerebral Organoid Kit | STEMCELL Technologies | 08570 | |
| Other | Millicell Cell Culture Insert | Sigma Aldrich | 32011202 | |
| Other | ACCUMAX cell dissociation reagent | Sigma Aldrich | A7089 | |
| Antibody | anti-RPL8 (Rabbit monoclonal) | Abcam | ab169538 | 1:200 |

*Continued on next page*

*Continued*

| Reagent type (species) or resource | Designation | Source or reference | Identifiers | Additional information |
|---|---|---|---|---|
| Antibody | anti-RPS10 (Rabbit monoclonal) | Abcam | ab151550 | 1:200 |
| Antibody | anti-RPS16 (Rabbit polyclonal) | Abcam | ab26159 | 1:200 |
| Antibody | anti-RPS26 (Rabbit polyclonal) | Thermo Fisher Scientific | PA5-65975 | 1:200 |
| Antibody | anti-S6 ribosomal protein (Mouse monoclonal) | Cell Signaling Technology | 5548 | 1:100 |
| Antibody | anti-SMI312 (Mouse monoclonal) | BioLegend | 837904 | 1:500 |
| Antibody | anti-MAP2 (Chicken polyclonal) | Abcam | ab5392 | 1:500 |
| Commercial assay or kit | CytoTune EmGFP Sendai Virus Fluorescence Reporter | Thermo Fisher Scientific | A16519 | |
| Chemical compound, drug | SiR-tubulin | Spirochrome | SC002 | |
| Other | Millicell Cell Culture Insert, 30 mm diameter, hydrophilic PTFE, 0.4 µm | Merck Millipore | PICM0RG50 | |
| Other | Disposable biopsy punch, 3.5 mm | Integra | 33–33 | |
| Other | Aspirator tube assemblies for calibrated microcapillary pipettes | Sigma-Aldrich | A5177-5EA | |
| Other | Capillary Tubes Glass, 50 µl volume, 100 mm length | Drummond Scientific | 1-000-0500 | |
| Other | P-2000 micropipette puller | Sutter Instrument | | |
| Other | EM grids, gold, 200 mesh, R 2/2 holey carbon | Quantifoil | | |
| Other | EM grids, gold, 200 mesh, R 3.5/1 holey carbon | Quantifoil | | |
| Software, algorithm | PEET | PMID:16917055 | | https://bio3d.colorado.edu/PEET/ |
| Software, algorithm | IMOD | PMID:8742726 | | http://bio3d.colorado.edu/imod/ |
| Software, algorithm | SerialEM | PMID:16182563 | | http://bio3d.colorado.edu/SerialEM/ |
| Software, algorithm | ec-CLEM plugin for Icy | PMID:28139674 | | |
| Software, algorithm | ImageJ | PMID:22930834 | | |
| Chemical compound, drug | poly-L-ornithine | Sigma Aldrich | P4957 | |
| Chemical compound, drug | Laminin | Sigma Aldrich | L2020 | |
| Chemical compound, drug | Fibronectin | Sigma Aldrich | F0895 | |

## Cell and cerebral organoid culture

The study employed H9 human embryonic stem cells (Wisconsin International Stem Cell Bank, Wicell Research Institute, WA09 cells), authenticated and confirmed to be mycoplasma-free by the provider, and approved for use in this project by the U.K. Stem Cell Bank Steering Committee. The H9 cells were cultured under feeder free conditions in StemFlex (Thermo Fisher Scientific, A3349401) on Matrigel (Corning, 356230) coated plates and passaged twice a week using 0.7 mM EDTA in sterile D-PBS without $Ca^{2+}$ and $Mg^{2+}$. Cerebral organoids were generated and grown using the STEMdiff Cerebral Organoid Kit (Stem Cell Technologies, 08570) according to manufacturer's guidelines. The HeLa cell line used carried a stably integrated doxycycline-inducible Fsp27-EGFP construct and was authenticated by PCR-single-locus-technology (Eurofins). HeLa cells were grown as an adherent culture in a

high-glucose DMEM media containing GlutaMAX (Thermo Fisher Scientific, 31996). The media was further supplemented with 10 % Tet-approved heat-inactivated FBS (Pan Biotech p30-3602), 10 mM HEPES pH 7.2, 0.2 mg/ml Hygromycin B (Invitrogen, 10687010) and 1 x non-essential amino acids solution (Thermo Fisher Scientific, 11140050). Cell lines were routinely tested for mycoplasma infection using the MycoAlert mycoplasma detection kit (Lonza).

## Plasmid construct generation

The study made use of the Sleeping Beauty transposon system. The constructs pCAGEN-SB100X, pT2-CAG-fGFP plasmid (Addgene #108714) and pT2-CAG-fFusionRed were previously used and described in other studies (*Lancaster et al., 2017*; *Giandomenico et al., 2019*). EGFP-E-Syt1 was a gift from Pietro De Camilli (Addgene plasmid # 66830) (*Giordano et al., 2013*). The construct pT2-CAG-GFP-E-Syt1 was generated by restriction digestion of the EGFP-E-Syt1 (Addgene #66830) and the pT2-CAG-fGFP plasmids with AgeI and MluI. The fragment encoding GFP-E-Syt1-SV40 PolyA was then ligated into the pT2-CAG backbone using T4 ligase. pcDNA3-hL1 was a gift from Fritz Rathjen (Addgene plasmid # 89411). The construct pT2-CAG-hL1CAM-GFP was generated by Gibson assembly on the fragments here described; the pT2-CAG-fGFP vector was linearised by restriction digestion with EcoRI and MluI, the fragment encoding the hL1CAM ORF was PCR amplified from the pcDNA3-hL1 plasmid (Addgene #89411) with primers Fwd: 5'-AACGTGCTGGTTATTGTGCTGTCT CATCATTTTGGCAAAGAAAGATGGTCGTGGCGCT-3' and Rev: 5'-TACAAGAAAGCTGGGTACCG TTCTAGGGCCACGGCAGGG-3', the fragment encoding EGFP-SV40 PolyA was PCR amplified from the pcDNA3-hL1 plasmid (Addgene #89411) with primers Fwd: 5'- ACCCTGCCGTGGCCCTAGAA CGGTACCCAGCTTTCTTGTACA-3' and Rev: 5'- CAGCCGGGGCCACTCTCATCAGGAGGGTTCAG CTTACTCAAGAGGCTCGAGTGCAACTATG-3'. The Q5 High-Fidelity 2 x Master Mix (New England Biolabs, M0492S) was used for PCR. The ligation and Gibson assembly products were transformed in TOP10 chemically competent *E. coli*. Plasmids recovered from bacterial colonies were screened by restriction digestion and the correct products were verified by sequencing.

## Construct expression in organoids

Organoids were electroporated with expression constructs as previously described (*Lancaster et al., 2013*; *Lancaster et al., 2017*; *Giandomenico et al., 2019*). Briefly, a total of 5 µl of a 1 µg/µl plasmid solution (750 ng/µl transposon donor plasmid and 250 ng/µl pCAGEN-SB100X) was injected into the ventricles of 45–55 day-old organoids and electroporated using the BTX Gemini X2 HT Electroporation System (BTX, 452008) and 5 mm gap petri dish platinum electrode kit to deliver five square-wave 1 ms pulses of 80 V amplitude with 1 s inter-pulse intervals. Approximately one week after electroporation the organoids were prepared for ALI culture.

## ALI-CO preparation

Cerebral organoids aged 45–60 days were prepared for ALI culture as previously described (*Giandomenico et al., 2019*). In brief, organoids were embedded in 3 % low-gelling temperature agarose (Sigma-Aldrich, A9414) in HBSS without $Ca^{2+}$ and $Mg^{2+}$ (Thermo Fisher Scientific, 14175095) and sectioned into 300 µm-thick slices on a Leica VT1000 S Vibrating blade microtome. All surrounding agarose was removed from the tissue and 2–3 tissue slices were positioned on each Millicell Cell Culture Insert (Merck Millipore, PICM0RG50) using No.22 scalpels (Swann-Morton, 0508). The slices were incubated for 1–2 hr in SSSC medium (0.5 % glucose, 10 % FBS and 1 x Anti-Anti in high glucose DMEM supplemented with Glutamax) and cultured long-term in SFSC medium (1 x B27, 0.5 % glucose, 1 x Glutamax and 1 x Anti-Anti in Neurobasal medium) with daily half-media changes.

## Organoid dissociation and neuronal culture

Mature organoids aged between 50 and 80 days were dissociated using ACCUMAX cell dissociation reagent (Sigma Aldrich, A7089) supplemented with 400 µg/ml DNAse I. For each organoid dissociated, 0.5 ml of dissociation solution were used. Organoids were resuspended in dissociation solution and subject to 4 × 5 min incubation steps in an incubator at 37 ° C; after the first 5 min the organoids were resuspended by flicking the tube, after 10 min the organoids displayed a fluffy appearance and were pipetted up and down once, then broken into cell clumps. After 15 min the cell clumps were

resuspended by pipetting up and down 3–5 times, and then 10 more times after an additional 5 min incubation. Dissociation was stopped by addition of an equal volume of maturation medium (Stem Cell Technologies, 08570). The cell suspension was passed through a 70 µm nylon cell strainer (Corning, 352350). A small aliquot was taken for a live cell count and the remaining cell suspension was spun down at 300 x g for 5 min. The cell pellet was resuspended in SFSC medium and 50,000 cells were seeded into each well of 8 well Lab-Tek II glass chamber slides (Nunc, 154534) for immunofluorescence preparation. Prior to cell seeding the imaging slide was coated with poly-L-ornithine, Laminin and Fibronectin – for details on the coating protocol refer to the section 'Electron cryo-microscopy (cryo-EM) sample preparation' of the Materials and Methods.

## Immunofluorescence sample preparation

Dissociated neurons were fixed in 4 % PFA for 10 min at room temperature and incubated in permeabilization buffer (4 % donkey serum and 0.25 % Triton-X in PBS without $Ca^{2+}$ and $Mg^{2+}$) for one hour at room temperature prior to overnight staining with primary antibodies in blocking buffer (4 % donkey serum and 0.1 % Triton-X in PBS without $Ca^{2+}$ and $Mg^{2+}$) at room temperature. Antibodies used in this study with the corresponding dilution factor were: rabbit anti-RPL8 (Abcam, ab169538, 1:200), rabbit anti-RPS10 (Abcam, ab151550, 1:200), rabbit anti-RPS16 (Abcam, ab26159, 1:200), rabbit anti-RPS26 (Thermo Fisher Scientific, PA5-65975, 1:200), Alexa Fluor 647 conjugate mouse anti-S6 ribosomal protein (Cell Signaling Technology, 5548, 1:100), mouse anti-SMI312 (BioLegend, 837904, 1:500), chicken anti-MAP2 (Abcam, ab5392, 1:500). The next day, the slides were washed three times in PBS, followed by a 1 hr incubation at room temperature with 405, 568, and 647 Alexa Fluor conjugate secondary antibodies diluted 1:500 in blocking buffer. After secondary antibody staining, the slides were washed three times in PBS and the coverslips were mounted using ProLong Diamond antifade mountant (Thermo Fisher Scientific, P36961).

## EmGFP Sendai virus transduction and SiR-tubulin labeling

Dissociated neuronal cultures used for immunofluorescence staining of ribosomal subunits were fed with 200 µl of SFSC medium supplemented with CytoTune EmGFP Sendai Fluorescence Reporter (Thermo Fisher Scientific, A16519, $8.1 \times 10^7$ CIU/ml) diluted 1:200. After 3–4 days, the cells started displaying EmGFP signal. Approximately 2 weeks after dissociation, cultures produced thin EmGFP⁺ axons and the cultures were fixed for analysis. SiR-tubulin (*Lukinavičius et al., 2014*) was reconstituted in sterile DMSO to a concentration of 1 mM. For staining of ALI-COs SiR-tubulin was diluted to a final concentration of 1 µM in SFSC medium and applied dropwise to the top of the slice using a controlled oral-suction pipetting apparatus and care was taken not to disturb the grids. After approximately 1 hr at 37 °C and 5 % $CO_2$ samples were ready for imaging.

## Fluorescence image acquisition and analysis

Widefield fluorescence images were acquired on a Nikon ECLIPSE Ti2 system at 10 x (0.3 NA) and 20 x (0.75 NA) magnification and on an EVOS FL inverted microscope (Thermo Fisher Scientific). Confocal images of SiR-tubulin stained organoids were acquired on a Zeiss LSM 710 upright system at 10 x (0.3 NA) magnification. The time course of fGFP⁺ axon growth on grids was acquired on a Zeiss LSM 780 confocal microscope using a 10 x (0.3 NA) objective and a pixel size of 830 nm. Samples were incubated at 37 °C and 5 % $CO_2$ in 35 mm Easy-Grip tissue culture dishes (Corning, 353001). The microscope objective was aligned to the center of the grid and 4 × 4 tiled-images were acquired approximately every 12 hr. For measurement of axon growth rates, live FM movies of ALI-COs were acquired on a Zeiss LSM 710 and Zeiss LSM 780 inverted microscope using a 10 x (0.3 NA) objective and a pixel size of 1.384 µm. Samples were incubated at 37 °C and 5 % $CO_2$ in 35 mm Easy-Grip tissue culture dishes (Corning, 353001) and images were acquired every 12 min. The manual tracking plugin in ImageJ was used to track the position of individual growth cones throughout the movie frames (Fabrice Cordelires, https://imagej.nih.gov/ij/plugins/track/track.html). The time interval was set to 12 min and the x/y calibration to 1.3837 µm. For immunofluorescence analysis of ribosomal subunit distribution in axons and dendrites, dissociated human neurons were imaged on a Zeiss LSM 780 confocal microscope at 60 x (1.4 NA oil) magnification. The main criterion for fluorescence image acquisition was the presence of both MAP2⁺/SMI312⁻ dendrites and MAP2⁻/SMI312⁺ axons within the field of view. ImageJ was used for analysis and the mean gray value of the ribosomal protein of interest

along the length of two axons and two dendrites per image were measured. Axons were identified as SMI312$^+$/MAP2$^-$ or, due to antibody incompatibility, in the case of S6 as GFP$^+$/MAP2$^-$ processes. Dendrites were identified as MAP2$^+$ processes. The ribosomal proteins imaged include: RPL8, RPS10, RPS16, RPS26, and S6. For each of these targets, the average mean gray value was calculated across axonal and dendritic segments and a Mann-Whitney unpaired two-tailed test was used for statistical comparison between the two groups.

## Electron cryo-microscopy (Cryo-EM) sample preparation

After 4–7 days, the ALI-COs started to display escaping processes and were thus considered ready for grid placement. Quantifoil R2/2 or R3.5/1 200 mesh Au grids with carbon film (Quantifoil) were coated with 0.01 % poly-L-ornithine solution (Sigma Aldrich, P4957) overnight at 4 °C. The next day, the grids were further coated with a solution of 10–20 µg/ml Laminin (Sigma Aldrich, L2020) and 0.001 % Fibronectin (Sigma Aldrich, F0895) in ultrapure water at room temperature for 4 hr. Organoid sections were inspected on an EVOS FL inverted microscope (Thermo Fisher Scientific) by brightfield or GFP fluorescence. The grids were placed at sites where single escaping processes could be seen by brightfield or near the fluorescent foci. Cryo-EM grids were blotted with Whatman filter paper grade 1 (GE Healthcare) and placed in direct contact with the edge of the organoid section. Importantly, the edge of the grid was juxtaposed to that of the organoid section, but not covered by it. Growth of the processes could be monitored daily based on GFP fluorescence. In some experiments, after approximately 2 weeks, SiR-tubulin (Spirochrome, CY-SC002) was applied dropwise on top of the grids to visualize all axon tracts. The grids were deemed ready for freezing earliest after 10–14 days, or once the axons reached an area within approximately five grid squares distance from the grid center. Immediately prior to plunge-freezing, the grids were hydrated by applying media dropwise using a controlled oral-suction pipetting apparatus and glass capillaries. This step was crucial to reduce desiccation of neuronal processes during blotting of the EM grids. Tracts on EM grids were detached from their cell bodies using a 3.5 mm disposable biopsy punch (Integra, 33–33) and immediately collected with an L5 clamp style thin-tip tweezer (Dumont, 72882-D), then backside blotted for 5–10 s with Whatman filter paper grade 1 (GE Healthcare) and vitrified in liquid ethane using an in-house built manual plunger. To minimize the time between detachment and cryo-fixation to less than 20 s, the procedure was carried out sequentially by two experimenters that worked next to one another; the first experimenter detached the grids from the organoid slice and handed the grids directly to the second experimenter who performed the blotting and plunge freezing.

## Cryo-fluorescence microscopy (Cryo-FM)

The grids were screened for ice thickness and fluorescent signals within axon tracts by cryo-FM with the Leica EM cryo-CLEM system. The system was equipped with a HCX PL APO 50 x (0.9 NA) cryo-objective (Leica Microsystems), an Orca Flash 4.0 V2 SCMOS camera (Hamamatsu Photonics), a Sola Light Engine (Lumencor), a L5 filter (Leica) for detection of GFP and a Y5 filter (Leica) for the detection of the SiR-tubulin stain. The microscope stage was cooled to –195 °C and the room was humidity controlled (below 25 %). A 2.0 × 2.0 mm montage of each grid was taken of the green (1 s, 30 % intensity), brightfield channel (30 ms, 70), and optionally of the far-red channel (1 s, 30 % intensity). Individual z-stacks of grid squares of interest were acquired in 1 µm steps to cover the full range of fluorescent signals. Correlation of fluorescent axon tracts on cryo-EM grid square maps was done in Icy using the ec-CLEM plugin (*Paul-Gilloteaux et al., 2017*) using landmark features and carbon film holes in cryo-FM and cryo-EM images.

## Preparation and focused ion beam (FIB) milling of control HeLa cells

Control HeLa cell samples were prepared as described in *Ader et al., 2019*. In short, HeLa cells were grown for 24 hr on holey carbon film Au grids (200 mesh, R2/2, Quantifoil), fed with oleic acid and induced with doxycycline for Fsp27-EGFP expression for an unrelated project. Sixteen hr post-induction, HeLa cells were stained for 1 hr with LipidTOX Deep Red dye (Thermo Fisher Scientific, H34477). Subsequently, grids were manually backside blotted with Whatman filter paper grade 1 and immediately vitrified in liquid ethane using an in-house built manual plunger. Thin lamellae were generated by cryo-FIB milling performed with a Scios DualBeam FIB/SEM (FEI) equipped with a Quorum stage (Quorum, PP3010T) in a procedure similar to the one described in *Schaffer et al., 2015*. Prior to

milling, grids were coated with organometallic platinum using a gas injection system for 30 s at 13 mm working distance and 25° stage tilt. The electron beam was used for locating the cells of interest at 5 kV and  13 pA beam current and for imaging to check progression of milling at 2 kV and  13 pA beam current. Milling was performed with stepwise reduction of the ion beam current (from 30 kV, 1 nA to 16 kV, 23 pA) while changing the stage tilt as described (*Hoffmann et al., 2019*). The lamellae with a 10° pre-tilt were milled to a final thickness below 300 nm.

## Electron cryo-microscopy (Cryo-EM)

Cryo-EM grids were screened on a Tecnai T12 (FEI) with an Orius camera or a Tecnai F20 (FEI) with a Falcon2 detector (FEI) by mapping the central parts of the grids using SerialEM (*Mastronarde, 2005*) at pixel sizes of 132 nm or 87 nm, respectively. The preservation of axon tracts on individual grids squares was examined on images acquired at pixel sizes of 6.3 nm or 6.0 nm, respectively. Cryo-ET data acquisition was done using SerialEM on a Titan Krios microscope (Thermo Fisher) equipped with a Quantum energy filter and a K2 direct electron detector (Gatan) operated in counting mode. Montaged images of the central part of the grid were acquired in linear mode with 171 nm pixel size. Montages of individual grid squares with axon tracts or lamellae of HeLa cells were taken with 5.1 nm pixel size. These montages were used for correlation to fluorescent axon tracts, based on landmark features using the ec-CLEM plugin (*Paul-Gilloteaux et al., 2017*) within the Icy software. Tilt series were acquired at areas of interest in low-dose mode from 0° to ±60° using a grouped dose-symmetric tilt scheme with 1° increment, a group size of 4 (*Hagen et al., 2017*), and a pixel size of 3.5 Å or 3.7 Å; both for the axon tracts and the control HeLa cells. The target dose rate was kept around 4 e$^-$/px/s on the detector. The energy filter slit width was set to 20 eV. Tilt images were acquired as three or four frames with approximately 1 e$^-$/A$^2$ dose per tilt image. The nominal defocus for all tilt series was set to –5 μm. The frames of tilt series images were aligned with IMOD alignframes. The tilt series were aligned in IMOD using patch tracking and then reconstructed at a pixel size of 7.1 Å or 7.4 Å as back-projection tomograms with SIRT-like filter corresponding to 10 iterations (*Kremer et al., 1996*; *Mastronarde, 1997*). To improve visibility for representation in figures, gaussian filtering was applied to the shown tomographic slices. For analyzing the microtubule polarity by subtomogram averaging, the contrast transfer function was estimated and corrected for by phase flipping in IMOD, and the tomograms were reconstructed by unfiltered backprojection at 7.1 Å or 7.4 Å pixel size. Four of the 9 HeLa tomograms used here have been analyzed and published before (*Ader et al., 2019* and EMD-4491).

## Image processing and analysis

The segmentation model shown in *Figure 2C* and in *Figure 2—video 2* was generated using Amira (Thermo Fisher Scientific) and IMOD (*Kremer et al., 1996*). Membrane surfaces were segmented manually, followed by extensive smoothening and simplification. Microtubules and actin filaments were first modeled as tubes in IMOD and then imported as tubular volumes into Amira. Within the segmented volumes of individual membrane objects and cytoskeleton objects, grey value-thresholding was used for a second segmentation step to eventually depict only high-density voxels within the objects of interest. Note that the segmentation model is inverted along the z-axis relative to the original tomogram. *Figure 2—video 1* was generated by first selecting a region of interest in a cryo-ET slice, which is presented in the final movie frame, followed by identifying and aligning images obtained from the previous successive imaging steps of fluorescent live imaging, cryo-FM, and electron tomography. Each acquired image was then used to generate a stack of progressive magnification into the region of interest using an ImageJ macro (zoom_movie_ImageJ_v2) written by Eugene Katrukha and provided through GitHub Gist (*Katrukha, 2021*). All stacks were then converted and concatenated into a single movie.

The membrane distance at cell-cell contact sites was measured for a total of 34 contacts, in which the plasma membranes were perpendicular to the x,y plane of the tomogram and contacting each other over a distance of at least 0.5 μm. 5–17 measurements, at a spacing of approximately 100 nm, were taken between the two extracellular leaflets of the plasma membrane bilayers for each contact area. Significance was tested with unpaired t test in Graphpad Prism.

The occurrence of ER-plasma membrane contact sites was assessed by visual inspection. Wherever the ER was found close to plasma membrane oriented perpendicular to the x,y plane of the tomogram, the distance between the two membranes was measured in IMOD and when it was less than

30 nm, the instance was counted as a membrane contact site. A large part of the plasma membrane however is oriented roughly parallel to the x,y plane of the tomogram and thus not directly visible because of the missing wedge. In these cases, we considered the boundaries between visible cellular material and the void in the top and bottom part of the tomograms to correspond to the surface of the cell. ER which was clearly recognizable less than 30 nm away from this boundary was considered an ER-plasma membrane contact site. Depending on the orientation of the ER membrane, however, the missing wedge affected its visibility as well and rendered the estimation of the distance to the plasma membrane impossible. Therefore, this approach only detects most obvious ER-plasma membrane contact sites.

The microtubule directionality was determined from the radial tilt of protofilaments in averages from individual microtubules. Points along individual microtubules were clicked along their center to create a contour. Overlapping subvolumes were extracted along each contour and averaged together using a cylindrical mask and one individual subtomogram as a reference in PEET (*Nicastro et al., 2006*; *Heumann et al., 2011*). Microtubule directionality was determined from axial views of individual subtomogram averages (*Sosa and Chrétien, 1998*). The number of protofilaments for individual microtubules was determined from axial views of subtomogram averages only when the quality of subtomogram averaging resulted in clear separation of the individual protofilaments. In some cases, analysis of subtomogram averaging was not possible due to the microtubule orientation and the anisotropic resolution of the tomograms, the signal-to-noise ratio of the final tomogram or the proximity to other microtubules within larger bundles. A subtomogram average was generated from overlapping subvolumes of 16 individual 13 protofilament microtubules using a cylindrical mask and the average of an individual microtubule average as a reference in PEET. Fourier shell correlation (FSC) was calculated in PEET using the calcFSC function. The FSC curve was plotted in Graphpad Prism using the resulting values, the frequency shells and the pixel size. A line profile along an individual protofilament of the microtubule average was generated in ImageJ and visualized using Graphpad Prism.

To measure the plasma membrane surface area of axon shafts, the plasma membrane boundaries of individual axons were segmented in tomograms at bin 10 that were rotated around the x-axis (to see x,z virtual slices). As explained for the analysis of membrane contact site occurrence, due to the missing wedge, the plasma membrane parallel to the x,y plane of the tomogram is not directly visible. However, the reconstructed tomographic volume contains the whole thickness of the cell, hence the boundary between cellular material and void corresponds to the surface of the cell. The surface of the cell is recognizable in the binned x,z virtual slices, and was therefore used as a proxy for the plasma membrane. Axon segments were only analyzed if they were fully contained throughout the tomogram volume. Segmentations and surface area measurements were performed in IMOD. Each measurement was normalized to the individual axon length.

The ER diameter of the thinnest ER tubules contained in the axon and HeLa datasets were measured in IMOD as the distance between two points on the cytosolic edge of the membrane of each ER tubule. Significance was tested with Welch's t test in Graphpad Prism.

Ribosome-like particles were identified visually on the basis of their size, shape, and contrast in the axon tomograms, in reference to other tomograms of eukaryotic cells that contained polysomes and monosomes. Vesicle and ribosome-like particle diameters were measured along their longest axis, in tomographic slices in IMOD. Multivesicular bodies and organelles larger than 200 nm in diameter were excluded from the vesicle analysis. To better visualize the topographic distribution of the ribosome-like particles, example volumes of 0.05 µm$^3$ from cytosolic areas of the tomograms are shown in *Figure 4a*, in which the position of each ribosome-like particle is illustrated by a sphere with 30 nm diameter. 'Other processes' comprised four tomograms containing high numbers of ribosomes. These tomograms were acquired at the edge of the EM grid, which is in close proximity to the cell bodies of the organoid slice and part of isolated processes rather than bundles. Additionally, two of these tomograms were acquired at the ending of a process. For these reasons, we interpreted them as representing cellular processes other than axonal tracts, excluded these four tomograms from the ribosome analysis on axon shafts and instead classified them as 'other processes'. The number of ribosome-like particles found within each tomogram was normalized to the tomographic volume. The tomographic volume was estimated from the x,y dimensions of tomograms. The z dimensions were estimated at the tomogram center. Using Avogadro's number, the average molar concentration of

ribosome-like particles within the tomograms of the axon tracts was calculated. Due to the permissive assignment and quantification accounting for their scarcity, it cannot be excluded that some of the ribosome-like particles identified in the axon shaft may actually correspond to other protein complexes. Thus, the calculated concentration in axon shafts is likely an overestimate.

## Acknowledgements

The authors thank members of the Kukulski and Lancaster labs for helpful comments and discussion, the light microscopy and EM facilities of the MRC Laboratory of Molecular Biology for support during data acquisition, Koini Lim for the HeLa cell line, Paul Donlin-Asp, Erin M Schuman and Benoît Zuber for critical reading of the manuscript. Work in the Lancaster lab is supported by the Medical Research Council (MC_UP_1201/9) and the European Research Council (ERC STG 757710). Work in the Kukulski lab was supported by the Medical Research Council (MC_UP_1201/8) and the University of Bern. MRW was supported in part by the Natural Sciences and Engineering Research Council (NSERC) of Canada (PGSD).

## Additional information

### Competing interests

Madeline A Lancaster: MAL is an inventor on several patents related to cerebral organoids, is co-founder and scientific advisory board member of a:head bio, and on the scientific advisory board of the Roche Institute for Translational Bioengineering. The other authors declare that no competing interests exist.

### Funding

| Funder | Grant reference number | Author |
|---|---|---|
| Medical Research Council | MC_UP_1201/9 | Madeline Lancaster |
| European Research Council | ERC STG 757710 | Madeline Lancaster |
| Medical Research Council | MC_UP_1201/8 | Wanda Kukulski |
| Natural Sciences and Engineering Research Council of Canada | | Michael R Wozny |

The funders had no role in study design, data collection and interpretation, or the decision to submit the work for publication.

### Author contributions

Patrick C Hoffmann, Stefano L Giandomenico, Conceptualization, Formal analysis, Investigation, Methodology, Writing – original draft; Iva Ganeva, Michael R Wozny, Magdalena Sutcliffe, Investigation, Writing – review and editing; Madeline A Lancaster, Conceptualization, Formal analysis, Supervision, Writing – original draft, Funding acquisition, Writing – review and editing; Wanda Kukulski, Conceptualization, Formal analysis, Supervision, Writing – original draft

### Author ORCIDs

Patrick C Hoffmann http://orcid.org/0000-0003-3421-6363
Stefano L Giandomenico http://orcid.org/0000-0003-4235-8353
Iva Ganeva http://orcid.org/0000-0003-3221-2502
Michael R Wozny http://orcid.org/0000-0001-9359-3287
Magdalena Sutcliffe http://orcid.org/0000-0001-5853-2331
Madeline A Lancaster http://orcid.org/0000-0003-2324-8853
Wanda Kukulski http://orcid.org/0000-0002-2778-3936

### Decision letter and Author response

Decision letter https://doi.org/10.7554/eLife.70269.sa1

Author response https://doi.org/10.7554/eLife.70269.sa2

## Additional files

### Supplementary files

• Supplementary file 1. File names of the cryo-ET data deposited on EMPIAR (*Iudin et al., 2016*), and the corresponding numbers given to the respective tomogram snapshots in *Figure 2—figure supplements 2–4*.

• Transparent reporting form

### Data availability

Representative electron tomograms are deposited at the Electron Microscopy Data Bank (EMDB); entries EMD-13195 for fGFP, EMD-13196 for L1CAM-GFP and EMD-13197 for GFP-ESYT1. The corresponding full data sets of raw tilt series image frames and reconstructed electron tomograms are deposited at the Electron Microscopy Public Image Archive (EMPIAR); entries EMPIAR-10806 for fGFP, EMPIAR-10805 for L1CAM-GFP and EMPIAR-10804 for GFP-ESYT1 (see Supplementary file1 for file name correspondence to panels in Figure 2—figure supplements 2–4). Four out of 9 cryo-tomograms of HeLa cells used here for comparison, have been published before (Ader NR et al. eLife 2019, EMD-4491). All other data is available in the main text or the supplementary materials. H9 cells are available from WiCell under a material transfer agreement with WiCell.

The following dataset was generated:

| Author(s) | Year | Dataset title | Dataset URL | Database and Identifier |
|---|---|---|---|---|
| Hoffmann PC, Giandomenico SL, Ganeva I, Wonzy MR, Sutcliffe M, Lancaster MA, Kukulski W | 2021 | Cryo-ET of axons from human cerebral organoids, expressing membrane-targeted GFP | https://www.ebi.ac.uk/emdb/error/entry/EMD-13195 | Electron Microscopy Data Bank, EMD-13195 |
| Hoffmann PC, Giandomenico SL, Ganeva I, Wozny MR, Sutcliffe M, Lancaster MA, Kukulski W | 2021 | Cryo-ET of axons from human cerebral organoids, expressing L1CAM-GFP | https://www.ebi.ac.uk/emdb/error/entry/EMD-13196 | Electron Microscopy Data Bank, EMD-13196 |
| Hoffmann PC, Giandomenico SL, Ganeva I, Wozny MR, Sutcliffe M, Lancaster MA, Kukulski W | 2021 | Cryo-ET of axons grown from human cerebral organoids, expressing ESYT1-GFP | https://www.ebi.ac.uk/emdb/error/entry/EMD-13197 | Electron Microscopy Data Bank, EMD-13197 |
| Hoffmann PC, Giandomenico SL, Ganeva I, Wozny MR, Sutcliffe M, Lancaster MA, Kukulski W | 2021 | electron cryo-tomograms of axons from human cerebral organoids, expressing membrane-targeted GFP | https://www.ebi.ac.uk/pdbe/emdb/empiar/entry/10806/ | Electron Microscopy Public Image Archive, EMPIAR-10806 |
| Hoffmann PC, Giandomenico SL, Ganeva I, Wozny MR, Sutcliffe M, Lancaster MA, Kukulski W | 2021 | electron cryo-tomograms of axons from human cerebral organoids, expressing L1CAM-GFP | https://www.ebi.ac.uk/pdbe/emdb/empiar/entry/10805/ | Electron Microscopy Public Image Archive, EMPIAR-10805 |
| Hoffmann PC, Giandomenico SL, Ganeva I, Wozny MR, Sutcliffe M, Lancaster MA, Kukulski W | 2021 | electron cryo-tomograms of axons from human cerebral organoids, expressing GFP-ESYT1 | https://www.ebi.ac.uk/pdbe/emdb/empiar/entry/10804/ | Electron Microscopy Public Image Archive, EMPIAR-10804 |

The following previously published datasets were used:

| Author(s) | Year | Dataset title | Dataset URL | Database and Identifier |
|---|---|---|---|---|
| Ader NR, Hoffmann PC, Ganeva I, Borgeaud AC, Wang C, Youle RJ, Kukulski W | 2019 | cryo-ET of cryo-FIB milled HeLa cell | https://www.ebi.ac.uk/emdb/error/entry/EMD-4491. | Electron Microscopy Data Bank, EMD-4491 |

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
