## [Decision Letter]

**Acceptance summary:**

This paper will be of interest both to neuroscientists and those developing cryo-electron tomography methods. The authors present an exciting new technique for studying human axons with cryo-tomography, based on extending axons from organoids on to EM grids.

**Decision letter after peer review:**

Thank you for submitting your article "Electron cryo-tomography reveals the molecular architecture of growing axons in human brain organoids" for consideration by *eLife*. Your article has been reviewed by 3 peer reviewers, and the evaluation has been overseen by a Reviewing Editor and Anna Akhmanova as the Senior Editor. The reviewers have opted to remain anonymous.

Essential revisions:

1) Provide a thorough comparison with existing methods for in situ structure determination, in particular focussing on comparison with recent literature.

2) Tone down claims of 'molecular' achievements to match the level of insight that is actually provided in the paper. Reviewer 2 suggests to perform subtomogram averaging of ribosomes, this is quite easily done and it would provide evidence for the claimed potential of determining molecular structures with the methodology presented here, so it would be good to include if possible. Also, please include more controlled quantifications throughout, as suggested by reviewer 1.

3) Discuss in more depth how this methodology can be further expanded to systems different from neurons (i.e. other organoids, pathological systems).

4) Reviewer 1 provides a list of specific comments that should be considered in a revised submission.

*Reviewer #1 (Recommendations for the authors):*

1) Regarding the quantification of plasma membrane surface area. The authors present the measurements as absolute, when cryo-tomography data suffers from a missing wedge artefact that makes segmentation of all surfaces of objects highly challenging. How all the membrane surface was accurately segmented should be explained, or it should be discussed if the measurements are intended as best estimates rather than absolute.

2) L1CAM-GFP or ESYT1-GFP overexpression were interesting experiments, but quantitative evidence should be presented to support the conclusion of no differences with GFP wild-types.

3) The paper could benefit from some experimental expansion. Could it be quantified where ribosomes are located relative to other organelles? Although ER minimum dimensions are thin relative to other cells, what about the maximum and average dimensions and quantity? Can ER relationships with different organelles and membrane be quantified? Is there anything different in human neurons compared to rodent ones? Would it be possible to show more vesicle examples, or even quantify the size ranges? Are vesicles associated with particular structures, i.e microtubules?

*Reviewer #2 (Recommendations for the authors):*

The manuscript could be supplemented by a comparison with findings from other methods. I am not an expert on axon development, but one might think of data from recent FIB/SEM approaches or from conventional electron microscopy studies, and I am not thinking of the mid-1950s papers that have already been cited here. This would allow even the non-expert to properly appreciate the findings presented.

In addition, I think it would be appropriate to add some more information. e.g. about the time needed from punching to plunging (page 5. lines 114-116). A related question is how stable the tissue is after punching, or at what point one can expect structural changes.

I also think that the "molecular" can be scaled down a bit, i.e. page 5, line 126-146 the only molecular is "numerous intraluminal particles" that remain to be investigated here, everything else are easily identifiable structures.

Also, maybe the authors should rephrase their questions from the introduction or really try to answer them directly in the rest of the manuscript. Right now they are either no answers or very vague answers. I think a little extra work here would definitely strengthen the manuscript.

Regarding "ribosomes", I would add a rough subtomogram analysis, which should be straightforward enough.

*Reviewer #3 (Recommendations for the authors):*

Although the authors describe a new methodology in this manuscript, but it lacks sufficient novelty and impact for publication in *eLife*. It will be great if the authors can demonstrate that this technique can be applied to other cell types/tissues or under different physiological or pathological conditions. This will make the work more impactful and not limit it to studying cerebral organoids or neurons.

---

## [Author Response]

Essential revisions:1) Provide a thorough comparison with existing methods for in situ structure determination, in particular focussing on comparison with recent literature.

We now extensively discuss recent literature that describes the use of EM to study neuronal cells, in particular volume EM methods which have over the last years provided immense advances in understanding neuronal tissue organisation. Furthermore, we compare these methods to cryo-EM and we discuss recent advances of in situ structure determination, in particular in the context of neuronal cells. This additional context is now included in the introduction.

See response to Reviewer 2 for additional details.

2) Tone down claims of 'molecular' achievements to match the level of insight that is actually provided in the paper. Reviewer 2 suggests to perform subtomogram averaging of ribosomes, this is quite easily done and it would provide evidence for the claimed potential of determining molecular structures with the methodology presented here, so it would be good to include if possible. Also, please include more controlled quantifications throughout, as suggested by reviewer 1.

We have toned down the claims of molecular achievements, as requested, in all instances it was mentioned, as well modified the title, to reflect these changes.

We appreciate the suggestion to perform subtomogram averaging (STA) of ribosomes as a way of providing evidence for the potential of the methodology and, as Reviewer 2 suggests, as a way of confirming the identity of ribosome-like particles. This would be an excellent addition, but precisely because axons are so ribosome-sparse (we have only identified less than 30 ribosome-like particles in 31 cryo-ETs of axons) STA of so few putative ribosomes is unlikely to provide any appreciable improvement in resolution or reveal further details beyond the size and shape of the average particle. Hence it may tell us that the *average particle* is likely a ribosome, but it won’t confirm the identity of the individual particles included in the average, especially since classifications or sorting of the data set will not be possible with such a small number of particles. We have therefore addressed the two points meant to be addressed by this suggestion separately, as follows:

Firstly, to highlight the potential of the methodology to visualize molecular structures, we have included a more extensive analysis of the structure of microtubules. We found that the majority of microtubules in the growing axons consists of 13 protofilaments. We calculated a 3D subtomogram average of the microtubules with 13 protofilaments, which reveals the repetitive arrangement of individual tubulin molecules along the protofilaments. These results are shown in new Figure 3 —figure supplement 1. We believe that these results could form the basis for other researchers to perform a targeted structural analysis of specific features of interest on microtubules. To facilitate such potential follow-up work, we have deposited on EMPIAR the complete set of raw images from tilt series acquired on growing axons. In addition, this data also demonstrates the utility of the approach for gaining insight into molecular structure of other subcellular axonal structures.

Secondly, to assign ribosome-like particles more confidently, we assessed the sizes of individual particles. The diameters were on average 27 nm, ranging from 22 to 36 nm, in good agreement with the dimensions of human ribosomes viewed from various angles (Anger A.M. *et al.,* Nature 2013, EMD-5592). This new data is shown in Figure 4 —figure supplement 1B. We are therefore confident that the majority of the particles we counted are indeed ribosomes.

Importantly, we would like to point out that our conclusion is that growing axon shafts contain very few ribosomes. The potential that our counts overestimate the number of ribosome due to false positives does not endanger that conclusion. The concentrations of ribosomes we provide are meant to highlight the large difference in ribosome occurrence between axon shafts and HeLa cells. Our counts may overestimate the number of ribosomes in axon shafts and underestimate the numbers in HeLa cells (because there are so many, we might have missed some). In either case, however, the conclusion would in fact be more strongly supported rather than weakened.

In addition, we provide a gallery of all the ribosome-like particles that we have found in growing axon shafts, as well as a quantification of their localization (cytosolic vs. ER-bound) (Figure 4 —figure supplement 1A and C).

3) Discuss in more depth how this methodology can be further expanded to systems different from neurons (i.e. other organoids, pathological systems).

We have now included a paragraph in the discussion discussing the potential application of our methodology to other systems. For instance, we discuss that it would be possible to study other elongating tissue structures, such as the developing vasculature, or organoid-derived cell types that exhibit migratory patterns and that would otherwise be difficult to culture, such as neural crest cells.

4) Reviewer 1 provides a list of specific comments that should be considered in a revised submission.

Please see below for details on how we addressed the specific comments of Reviewer 1 as well as the other Reviewers.

Reviewer #1 (Recommendations for the authors):1) Regarding the quantification of plasma membrane surface area. The authors present the measurements as absolute, when cryo-tomography data suffers from a missing wedge artefact that makes segmentation of all surfaces of objects highly challenging. How all the membrane surface was accurately segmented should be explained, or it should be discussed if the measurements are intended as best estimates rather than absolute.

We thank the reviewer for raising this important point. Indeed, our measurements of plasma membrane surface area are estimates and not intended to be absolute values. It is correct that the segmentation of membrane areas that are oriented perpendicular to the optical axis is extremely difficult because membranes in that orientation are essentially not visible in tomographic reconstructions. However, the surface of the imaged cellular volume is visible in x,z or y,z-views of the tomograms. As the axons are imaged without any thinning, we assume that the visible surface of the imaged cellular volume is a good proxy for the position of the plasma membrane of the axon.

We have adjusted the relevant sentence in the main text to read:

“we segmented the plasma membrane in tomograms and estimated the surface area of individual axon shafts (Figure 3F, see also Materials and methods). Each micrometer of length on average corresponded to approximately 1.24 x 10^6^ nm^2^ of plasma membrane area (SD = 7.3 x 10^5^ nm^2^, N = 14) (Figure 3G).”

In the Materials and methods section, we provide a detailed description of the approach:

“To measure the plasma membrane surface area of axon shafts, the plasma membrane boundaries of individual axons were segmented in tomograms at bin 10 that were rotated around the x-axis (to see x,z virtual slices). As explained for the analysis of membrane contact site occurrence, due to the missing wedge, the plasma membrane parallel to the x,y plane of the tomogram is not directly visible. However, the reconstructed tomographic volume contains the whole thickness of the cell, hence the boundary between cellular material and void corresponds to the surface of the cell. The surface of the cell is recognizable in the binned x,z virtual slices, and was therefore used as a proxy for the plasma membrane.”

2) L1CAM-GFP or ESYT1-GFP overexpression were interesting experiments, but quantitative evidence should be presented to support the conclusion of no differences with GFP wild-types.

We have now included quantitative analyses, according to the anticipated function of the two overexpressed proteins. For axons from L1CAM-GFP overexpressing organoids, we measured whether the distance between the plasma membranes of adjacent axons is altered compared to axons from fGFP expressing organoids. These data are shown in Figure 2 —figure supplement 5. In axons from GFP-ESYT1 overexpressing organoids, we assessed the occurrence of ER-plasma membrane contact sites as compared to fGFP data. This analysis includes additional, representative examples of ER-plasma membrane contact sites in Figure 2 —figure supplement 6 and quantification reported in the results. In both cases, we found no difference. These results are described as follows:

“We compared cell-cell contacts formed between individual axons in tracts from ALI-CO slices expressing fGFP with tracts from ALI-CO slices overexpressing the cell adhesion molecule L1CAM. While the separation between plasma membranes of adjacent axons within tracts was remarkably narrow, we found no significant difference between cell-cell contacts from fGFP and L1CAM-GFP overexpressing ALI-COs (fGFP: mean = 5.63 nm, SD = 0.65 nm, N = 25 and L1CAM-GFP: mean = 5.29 nm, SD = 0.58 nm, N = 9) (Figure 2 —figure supplement 5). We also compared the occurrence of ER-plasma membrane contact sites in axons from ALI-CO slices expressing fGFP with axons from ALI-CO slices overexpressing the ER-plasma membrane contact site protein GFP-ESYT1 (Fernández-Busnadiego et al., 2015). The frequency with which we observed such contact sites was similar in both data sets (86% of fGFP tomograms (N = 22) and 90% of GFP-ESYT1 tomograms (N = 10) contained at least one instance in which the ER was within approximately 30 nm distance from the plasma membrane) (examples in Figure 2 —figure supplement 6). We cannot exclude that there may be subtle unanticipated phenotypes associated with other, specific structures, nor that more severe phenotypes could be masked by the mosaic nature of the bundles, with wild type axons possibly driving growth of axons expressing the transgene. Nonetheless, the intactness of the subcellular organization suggests that in principle our approach allows one to assess the impact of gene manipulation on cellular ultrastructure without technical knock-on effects.”

3) The paper could benefit from some experimental expansion. Could it be quantified where ribosomes are located relative to other organelles? Although ER minimum dimensions are thin relative to other cells, what about the maximum and average dimensions and quantity? Can ER relationships with different organelles and membrane be quantified? Is there anything different in human neurons compared to rodent ones? Would it be possible to show more vesicle examples, or even quantify the size ranges? Are vesicles associated with particular structures, i.e microtubules?

We thank the Reviewer for these suggestions. We have done several additional analyses.

– We have quantified whether ribosomes are cytosolic or bound to ER, and we included a gallery of all axonal ribosomes we observed within their subcellular environment (Figure 4 —figure supplement 1) (see response to essential revisions point 2).

– We have quantified the occurrence of ER-plasma membrane contact sites (see response to major point 2).

– We have done various analyses on vesicles, as suggested (Figure 2 —figure supplement 1). We provide an image gallery to show more example vesicles. We assessed whether vesicles were associated with microtubules, actin or whether vesicles appeared free in the cytosol. Finally, we measured the diameters of vesicles and compared the sizes of free vesicles and those associated with microtubules or actin. These results are described in an additional paragraph as follows:

“First, we investigated the localization of vesicles (Figure 2A and Figure 2—figure supplement 1A). While most vesicles appeared free in the cytosol, approximately 16% were associated with microtubules and 14% with actin (N=298, Figure 2—figure supplement 1B). The vesicles had mean diameters of about 50 nm (52.00 nm, SD = 19.42 nm, N = 200 for free vesicles; 50.92 nm, SD = 16.70 nm, N = 49 for MT associated vesicles, and 52.18 nm, SD = 17.84 nm, N = 41 for actin associated vesicles) (Figure 2 —figure supplement 1C). These sizes are similar to vesicles in axons of mouse dorsal root ganglia neurons (Foster et al., 2021b). Although we did not determine the origin and identity of the vesicles, their distribution suggests that at least a subset could correspond to secretory vesicles (Gumy et al., 2017).”

Reviewer #2 (Recommendations for the authors):The manuscript could be supplemented by a comparison with findings from other methods. I am not an expert on axon development, but one might think of data from recent FIB/SEM approaches or from conventional electron microscopy studies, and I am not thinking of the mid-1950s papers that have already been cited here. This would allow even the non-expert to properly appreciate the findings presented.

We agree with the Reviewer that recent EM developments, such as FIB-SEM imaging, have contributed immensely to understanding the architecture of neuronal cells. We have therefore expanded our introduction to discuss the advances made by various imaging approaches.

Furthermore, we discuss the concrete example of very narrow ER diameter measurements, which has been previously described by FIB-SEM imaging. Our measurements show that the ER is much narrower than previously found by FIB-SEM imaging (Terasaki et al., 2018). This does not mean the FIB-SEM measurements are wrong (it could be different in different cell systems), but the narrow diameters we measure could not have been accurately measured with the resolution of the FIB-SEM or any other EM method.

I also think that the "molecular" can be scaled down a bit, i.e. page 5, line 126-146 the only molecular is "numerous intraluminal particles" that remain to be investigated here, everything else are easily identifiable structures.

We have now replaced the term ‘molecular’ in all relevant instances, including the title. In the instance mentioned by the Reviewer, we have replaced ‘to molecular detail’ by ‘to a high level of detail’.

Also, maybe the authors should rephrase their questions from the introduction or really try to answer them directly in the rest of the manuscript. Right now they are either no answers or very vague answers. I think a little extra work here would definitely strengthen the manuscript.

We thank the Reviewer for this suggestion. We have now rephrased the introductory questions to better match the presented results.

“How are cellular compartments and the cytoskeleton organized within the rapidly extending axon? Does the subcellular organization provide the means to understand the supply of lipids and proteins, necessary for the increase in axon surface area during growth?”

Regarding "ribosomes", I would add a rough subtomogram analysis, which should be straightforward enough.

Please see our response to essential revisions point 2.

Reviewer #3 (Recommendations for the authors):Although the authors describe a new methodology in this manuscript, but it lacks sufficient novelty and impact for publication in eLife. It will be great if the authors can demonstrate that this technique can be applied to other cell types/tissues or under different physiological or pathological conditions. This will make the work more impactful and not limit it to studying cerebral organoids or neurons.

Please see our response to essential revisions point 3.